# PET-Assessed Metabolic Tumor Volume Across the Spectrum of Solid-Organ Malignancies: A Review of the Literature

**DOI:** 10.3390/biomedicines13010123

**Published:** 2025-01-07

**Authors:** Anusha Agarwal, Chase J. Wehrle, Sangeeta Satish, Paresh Mahajan, Suneel Kamath, Shlomo Koyfman, Wen Wee Ma, Maureen Linganna, Jamak Modaresi Esfeh, Charles Miller, David C. H. Kwon, Andrea Schlegel, Federico Aucejo

**Affiliations:** 1Department of Internal Medicine, Cleveland Clinic Foundation, Cleveland, OH 44195, USA; agarwaa7@ccf.org; 2Digestive Diseases and Surgery Institute, HPB and Transplant Surgery, Cleveland Clinic Foundation, Cleveland, OH 44195, USA; 3Department of Radiology, Nuclear Medicine Section, Cleveland Clinic Foundation, Cleveland, OH 44195, USA; 4Taussig Cancer Institute, GI Oncology Section, Cleveland Clinic Foundation, Cleveland, OH 44195, USA; 5Taussig Cancer Institute, Head & Neck Oncology Section, Cleveland Clinic Foundation, Cleveland, OH 44195, USA; 6Digestive Diseases and Surgery Institute, Hepatology Section, Cleveland Clinic Foundation, Cleveland, OH 44195, USA

**Keywords:** FDG-PET scan, nuclear medicine, metabolic tumor volume

## Abstract

Solid-organ malignancies represent a significant disease burden and remain one of the leading causes of death globally. In the past few decades, the rapid evolution of imaging modalities has shifted the paradigm towards image-based precision medicine, especially in the care of patients with solid-organ malignancies. Metabolic tumor volume (MTV) is one such semi-quantitative parameter obtained from positron emission tomography (PET) imaging with ^18^F-fluorodeoxyglucose (FDG) that has been shown to have significant implications in the clinical oncology setting. Across various solid tumor malignancies, including lung cancer, head and neck cancer, breast cancer, esophageal cancer, and colorectal cancer, the current literature has demonstrated an association between MTV and various clinical outcomes. MTV may be used in conjunction with several existing and established clinical parameters to help inform risk stratification and treatment strategies and predict outcomes in cancer. Optimizing such volumetric parameters is paramount for advancing efforts to advance cancer care for our patients. While such advancements are made, it is important to investigate and address the limitations of MTV, including variability in terms of measurement methods, a lack of standardized cut-off values, and the impact of inherent tumor heterogeneity. Despite these limitations, which can precipitate challenges in standardization, MTV as a prognostic factor has great potential and opens an avenue for the future integration of technology into an image-based precision medicine model of care for cancer patients. This article serves as a narrative review and explores the utility and limitations of PET-MTV in various settings of solid-organ malignancy.

## 1. Background/Introduction

Cancer continues to impose a high disease burden worldwide and remains the second leading cause of death in the United States [1]. In the last three decades, advancement in imaging modalities has prompted the evolution of image-based precision medicine, thereby drastically improving patient care and outcomes. Positron emission tomography (PET) imaging with ^18^F-fluorodeoxyglucose (FDG) is a significant prognostic and predictive factor for solid tumors that has been instrumental in this rapid change. In addition to qualitative tumor assessment, PET-FDG imaging studies also permit quantitative tumor assessment that confers significant clinical benefit. The mainstay of PET-FDG involves the evaluation of tumor burden through the assessment of tissue metabolic activity. Cells with high glucose metabolism, a characteristic of several solid-organ malignancies, will uptake FDG, which is a glucose analog. The accumulation of FDG within tumor cells often allows for precise tumor assessment, including specific metabolic activity [2].

Standardized uptake value (SUV), a PET-FDG quantitative measurement that objectively measures FDG uptake in lesions at a certain point in time, was historically applied in so-called radiomics assessment due to its inter-observer reliability and relative ease of calculation through available software. However, SUV measurements were limited by imaging techniques, including FDG uptake time and scanner resolution [2]. More recently, in the last two decades, volumetric indices such as metabolic tumor volume (MTV) and total lesion glycolysis (TLG) have gained traction in improving the prognostication of solid-organ malignancies [2,3]. MTV represents the volume of an FDG-avid tumor and TLG is a product of MTV and SUV_mean_.

In 1999, Larson et al. presented the concept of TLG in PET-FDG, demonstrating that this parameter may be used to assess chemotherapy treatment response [4]. This landmark study catalyzed a paradigm shift that supported the use of MTV and TLG in clinical practice for the management of solid tumors, especially non-small-cell lung cancer (NSCLC). Since then, there has been notable increase in use of PET-MTV for several solid-organ malignancies. The National Cancer Institute has provided recommendations on the use of PET-FDG, and specifically PET-MTV, to assess changes in tumor cell viability and assist in prognostication and monitoring treatment response [5]. To build upon the current body of literature, our article will consolidate and discuss the broad applications of PET-MTV in various solid-organ malignancies. We will further discuss the potential future integration of PET-MTV with other advanced methods of precision oncology. Our narrative review was conducted using articles derived from PubMed using keywords such as “PET-MTV”, “metabolic tumor volume”, and “PET-FDG”. The citations of manuscripts were also reviewed for potentially relevant articles. Two reviewers searched the literature independently to ensure thorough review.

## 2. Lung Cancer

Lung cancer is one of the first disease states for which the utility of PET-FDG for staging and monitoring therapeutic response was described. Since then, PET-FDG has become a widely accepted tool for lung cancer in practice due to robust data on the uses and limitations of this imaging modality. Broadly, lung cancer is thought of in two main categories: non-small-cell lung cancer (NSCLC), which accounts for about 80% of lung cancer, and small-cell lung cancer (SCLC) [6].

In NSCLC, studies have demonstrated that higher MTV measurements are associated with worse prognosis, including both recurrence after surgery and overall survival in resected and advanced cases. A metanalysis conducted in 2015, which pooled 13 studies assessing 1581 patients, revealed that patients with high MTV had a significantly worse prognosis with hazard ratios of 2.71 for adverse events and 2.31 for death [7]. Studies have also demonstrated that MTV, including pretreatment MTV assessment, can serve as an independent predictor for locoregional control and overall survival (OS) in NSCLC [7,8]. In the context of surgical management, several studies reveal that higher MTV values are linked to worse postsurgical outcomes, serving as significant prognostic factors to predict survival in surgical NSCLC patients [9,10]. Interestingly, in early-stage NSCLC, Hyun et al. demonstrated that higher MTV values are associated with an increased risk of recurrence and death after surgical resection, suggesting the role of postsurgical MTV in further guiding treatment response [11]. More recently, with the rapidly advancing field of immunotherapy, Tricarico et al. assessed the role of MTV in NSCLC patients undergoing treatment with immune checkpoint inhibitors and demonstrated that the semi-quantitative approach of MTV is an optimal prognostic measure that exceeded and complemented the PERCIST criteria [12].

Similar findings have been posited for SCLC. Studies have shown higher MTV values being associated with shorter OS and worse prognosis in SCLC, depicting MTV as an independent predictor of both tumor progression and death, even in patients undergoing chemoradiation [13,14,15]. The role of MTV in surgical decision making for SCLC is less well understood, but the aforementioned findings of the prognostic value of MTV in SCLC underscore the potential utility of MTV in guiding risk stratification for patients who are being evaluated for surgical resection in this aggressive malignancy as part of a multi-modal treatment approach.

It is important to recognize that while the utility of MTV appears promising in lung cancer, the heterogeneity of current evidence may impact its broad applicability. For instance, in NSCLC, most data are derived from retrospective review studies and meta-analyses that pool a limited number of studies comprising different patient populations. Only two adequately powered prospective studies, performed by Ohri et al. (a multi-center prospective study) [8] and Tricarico et al. (a single-center prospective study) [12], were found. Data for SCLC are even more limited. The only available and included study was in the form of a single-center retrospective review article. While internal validity was achieved in the context of utilizing a homogenous patient cohort in this study, external validity is questioned, as the differences in tumor characteristics based on regionality are well established in precision oncology. As such, the variable study designs employed, in the current literature, to assess the role of MTV in lung cancer suggests that larger, prospective studies are needed to develop robust data that can help standardize MTV cut-off values and be largely applicable across varying patient populations.

## 3. Head and Neck Cancer

Head and neck cancers (HNCs) are a group of heterogenous diseases that include malignancies of the oropharynx, hypopharynx, larynx, nasopharynx, and sinonasal tract. Histologically, HNCs are similar but variations in natural course and clinical disposition based on primary tumor location make for clinical heterogeneity [16]. Currently, the American Joint Commission on Cancer utilizes TNM staging as its main strategy for evaluation for treatment response and prognosis.

Initial studies exploring the utility of metabolic parameters such as MTV and TLG often exhibited great variation in terms of results, representative of the heterogenous nature of HNCs. An initial retrospective cohort study conducted by La et al. suggested that traditional morphometric tumor burden in HNCs found larger tumor volumes as measured by MRI correlate with inferior local control and overall survival at 5 years [17]. Several metanalyses, conducted since then, have posited that MTV serves as a good prognostic factor in HNC [18,19,20]. Pak et al. revealed, in a pooled analysis of 13 studies comprising 1180 patients (statistical heterogeneity of 0%), that high MTV was associated with a 3.06-fold higher risk of adverse events or a 3.51-fold higher risk of death when compared to patients with lower MTV [18]. Despite the variability that may exist for volumetric parameters due to heterogeneity, MTV does have predictive value and clinical utility in event-free survival and overall survival [18,19]. Work on HNC has demonstrated the utility of radiomic and serologic biomarkers including 40-Gene Expression Panel testing in prognostic stratification and even therapeutic selection, highlighting the potential role(s) for combinatorial approaches to precision oncology [21,22].

As previously alluded, the inherent heterogeneity of HNC produces great variability in the results of MTV values and cut-off, which, in turn, has also impacted the quality of evidence available on the utility of MTV in HNC. Even the metanalyses conducted by Pak et al. and Rijo-Cedeno et al. [18,19], which, respectively, pooled 13 studies and 18 studies, posit that heterogeneity within the tissue characteristics of HNC and the heterogeneity of study designs included in the metanalyses made it very challenging to propose standard MTV cut-offs that may help inform decision making. While statistically significant trends were found in MTV values and overall survival, robust data that account for such HNC tumor heterogeneity are needed to protocolize the use of MTV in HNC. With this promising yet limited data, more work is needed in developing a standardized approach in utilizing MTV as a prognosticator in HNC, including prospective trials establishing its utility, alone or in isolation, as a guide for therapeutic selection.

## 4. Breast Cancer

Breast cancer is the most common cancer in women and is one of the leading causes of death in the United States [23]. A multimodal approach that is focused on individual molecular and biologic morphology is required to help prognosticate patient outcomes due to the inherent heterogeneity that exists in this subset of malignancies [24]. For example, tumor grade and size, and the presence of hormonal receptors and human epidermal growth receptors 2 (HER2s) are crucial in predicting disease outcomes in patients, due to which solely relying on tumor morphology might not be optimal [24]. Historically, PET-FDG has proven to be a useful imaging modality for tumor staging, with SUV_max_ being widely utilized in clinical settings, both for local and distant disease. However, early studies demonstrate that SUV_max_ depicts tumor characteristics based on the greatest intensity of FDG uptake within the tumor, which may not reflect the intra-tumor heterogeneity sufficiently [25]. As such, MTV and other volumetric parameters have gained traction.

Like other solid tumor malignancies, a higher MTV value has been shown to be associated with poorer outcomes, such as a higher risk of recurrence and lower overall survival times. A meta-analysis conducted by Pak et al. demonstrated that a greater MTV value from the primary tumor in breast cancer is significantly associated with a higher risk of recurrence and progression [26]. Another study, carried out by Higuchi et al., revealed that low baseline MTV and a significant reduction in SUV after preoperative chemotherapy were associated with achieving pathologic complete response (pCR) [27]. MTV has also been shown to be an independent prognostic factor of OS in patients with distant metastatic disease at time of breast cancer diagnosis [28]. Similarly, in the context of neoadjuvant therapy, baseline MTV and its reduction rate after neoadjuvant therapy were significant independent prognostic factors for disease-free survival (DFS) [29].

When looking across different cancer types, MTV can serve as a valuable biomarker. In ER-positive/HER2-negative breast cancer, high baseline MTV is an independent negative predictor of recurrence-free survival (RFS) and OS [27,30]. Additionally, in triple-negative breast cancer (TNBC), high MTV is associated with poorer disease-free survival (DFS) and overall survival (OS) [27]. In HER2-positive breast cancer, changes in MTV have been linked to early metabolic response, which can help predict pCR [27].

As it stands, current data on the role of MTV in breast cancer are primarily derived from single-center retrospective review studies. All studies had several similar limitations, including sample size, heterogeneity in patient population (including variation in follow-up time, molecular and biologic tumor morphology, treatment modality (chemotherapy, radiation, surgery, hormonal therapy, immunotherapy), and the timing of imaging). Even in the metanalysis conducted by Pak et al., which pooled nine studies consisting of 975 patients, for the sub-analysis on prognostic value of MTV, only 133 patients from two retrospectives were eligible for use, limiting the applicability of the data [26]. Additionally, as mentioned previously, breast cancer also confers intra-tumor heterogeneity which has impacted the use of PET-FDG, though this has not been explicitly studied in MTV use. Such tumor and study design variations highlight the dire need for consistent experimental designs and future prospective trials addressing the aforementioned limitations to understand the true utility of MTV across different breast cancer subtypes. Overall, MTV has been helpful as a prognostic factor across breast cancer subtypes and treatment modalities.

## 5. Esophageal Cancer

Esophageal cancer (EC) is the sixth leading cause of cancer-related deaths worldwide [31,32]. It has been predicted that if the current incidence rate of esophageal cancer remains stable, the global burden of esophageal cancer will increase to approximately 957,000 new cases and 880,000 deaths by 2040 [32]. Histologically, esophageal cancer is mainly classified into two subtypes: esophageal squamous cell carcinoma (ESCC) and esophageal adenocarcinoma (EAC). Esophageal SCC, most prevalent in Eastern Europe and Asia, accounts for about 85% of EC cases while EAC, most prevalent in North America and Western Europe, comprises about 14% of EC cases [32]. Esophageal cancer is particularly challenging due to its aggressive nature and challenges with resectability, making improved precision approaches uniquely critical. Perhaps for this reason, the utility of MTV is highly advanced in esophageal malignancy.

Several studies have demonstrated that higher MTV correlates with prognosis as in other tumor types. For instance, Lemarignier et al. conducted a retrospective study and found that higher pretreatment MTV correlated with shorter DFS and OS in ESCC patients who are currently undergoing chemotherapy [33]. Shum et al. reports in their small retrospective review of 26 patients that, in the context of curative surgical resection, greater MTV was associated with inferior one-year OS compared to lower MTV values [34]. Comparably, another study depicted that total MTV was a significant independent prognostic factor for relapse-free survival (RFS) in patients with thoracic ESCC that is amenable to surgical resection, which would suggest that patients with higher total MTV are linked to having worse outcomes [34]. In terms of non-surgical management, MTV may aide in predicting local control in nonsurgical candidates, and greater MTV is associated with worse OS and DFS in patients treated with definitive chemoradiotherapy [35,36].

MTV may currently influence treatment options and clinical management in esophageal adenocarcinoma. Tamandl et al. demonstrated that MTV is an independent predictor of OS in patients with advanced adenocarcinoma [37]. The same study also showed that posttreatment MTV with neoadjuvant chemotherapy and its reduction ratio (MTV_RATIO_) are independent predictors of OS in patients undergoing neoadjuvant chemotherapy and resection for EAC [37]. Makino et al., correspondingly, found that changes in MTV can be used to assess treatment response: a greater than 60% reduction in MTV after preoperative chemotherapy was associated with histologic response and significantly better progression-free survival (PFS) [38]. Interestingly, studies that assessed demographic data as prognostic indicators found that while age and gender are considered in prognostic models, there is no current direct evidence that establishes a correlation between MTV, and age, gender, and ethnicity [39].

At the intersection of radiologic parameters and technology lies potential in utilizing MTV for staging purposes. In EC, clinicians utilize a multimodal approach for pretreatment staging, including PET computed tomography (PET-CT) and endoscopic ultrasound (EUS). PET-CT is useful in the evaluation of the metabolic and anatomic properties of a metabolically active primary tumor and distant metastases while EUS is critical in evaluating the depth of tumor invasion and local lymph node metastasis. Malik et al. conducted a prospective study that demonstrated that MTV may complement EUS in pretreatment staging and has potential for early use in guiding clinicians and patients in management decision making [40].

Despite the vast advancements in the utility of MTV in EC, it is important to be critical of the evidence. Most available studies are small single-center retrospective studies that assess a patient population at a tertiary care center, which questions the broad applicability of these data. Interestingly, while observed trends were significant, the authors of these studies highlighted the variation in MTV values between EAC and ESCC due to inherent histologic differences in tumor pathology. Additionally, PET imaging machine and treatment modality were consistently reported as contributors to the heterogeneity of the patient population despite being studied at the same institution. Thus, while these studies demonstrate the immense potential of MTV in both EAC and ESCC, far more than other solid-organ malignancies, current data are not sufficient to integrate MTV into practice-changing guidelines. Further prospective trials, preferably through international collaborations due to known differences in EAC and ESCC risk factors and prevalence based on region, are needed to protocolize use of MTV in EC.

## 6. Colorectal Cancer

Colorectal cancer (CRC) is the third most common cancer worldwide and the second most common cause of cancer-related death in the United States [41]. While the overall incidence of CRC has somewhat declined over the past decade, there has been a steady rise in CRC diagnosis in patients younger than 55 years [41]. There has also been a concurrent increase in mortality from CRC in patients younger than 50 years of age [41]. With the advent of earlier CRC screening, there have been significant strides in cancer risk reduction, but CRC remains a global health burden due to existing disparities in screening and care [42].

MTV has been shown to have significant prognostic implications in colorectal cancer. To start, greater MTV values are associated with poorer outcomes in metastatic CRC (mCRC). Woff et al. demonstrated that whole-body metabolically active MTV (WB-MATV) and early metabolic tumor response (mR) serve as strong independent predictors of OS and PFS regardless of treatment modality in mCRC [43]. Interestingly, high MTV has been linked to microsatellite instability (MSI). Owing to the absence of one or more mismatch repair (MMR) system gene, microsatellites become even more prone to replication errors, making MSI a useful biomarker in CRC management. MSI is used to help predict treatment response to immunotherapy. A retrospective study conducted by Liu et al. reported that the PET volumetric parameters obtained correlated with MSI, with MTV_50%_ having the greatest ability to discriminate high and low MSI [44]. These results, if validated, can have significant implications in guiding immunotherapy in CRC patients, which would only be traditionally indicated for MSI-high patients.

Additionally, there has been a growing interest in the use of MTV in colorectal liver metastasis (CRLM) management. Patients with high MTV often have worse outcomes, including a higher risk of adverse events or death, as compared to those with lower MTV in CRLM [45]. In select patients with CRLM, liver transplantation (LT) remains a viable treatment option. A landmark Norwegian study demonstrated that MTV is a valuable metric that can be used to holistically assess disease burden within the liver, specifically showing that MTV measurement is highly predictive of long-term OS and DFS after LT in the management of CRLM [46,47,48]. This work was subsequently validated by Wehrle et al. in a recent multi-center study, and PET-MTV is now proposed for incorporation into transplant selection guidelines [48]. Consequently, the same group has built on this work and the utility of circulating tumor DNA (ctDNA) in this context, describing the “Cleveland Protocol” for the pretransplant management of patients with CRLM prior to liver transplantation, including the use of PET-MTV to guide the process of down-staging and the timing of curative-intent liver transplantation [49,50,51].

Similarly to the current state of affairs in the aforementioned solid-organ malignancies, the evidence that exists on the possible role of MTV in colorectal cancer mostly arise from single-center retrospective reviews. The majority of the studies here were again limited by small sample sizes, temporal bias given the evolution and advancement of treatment modalities in CRC, and tumor-specific genetic profiles, giving rise to heterogeneity even in single-center studies. As highlighted in the earlier work on relationship between MSI and MTV, when assessing response to immunotherapy, changes in the genetic makeup of the CRC tumor profile may play a role in impacting the cut-off values that could be used for MTV-guided prognostication. As such, prospective studies are again needed to further validate these findings and intentionally study the impact of intra-tumor and inter-tumor variability on MTV. Importantly for the CRLM studies that explored MTV in LT, it is imperative to recognize that while the well-characterized multi-center studies cited had limited sample sizes, prospective studies may be difficult to conduct due to the ethical considerations of designing prospective trials in liver transplant-eligible patients.

## 7. Hepatocellular Carcinoma

Hepatocellular carcinoma (HCC) is the most common primary liver malignancy and the fifth leading cause of cancer-related death [52,53]. HCC is further increasing in prevalence due to the continuing rise in non-alcoholic and alcoholic liver diseases [52,53]. Due to its frequent co-presentation with cirrhosis, the treatment of choice for HCC is liver transplantation when possible. The first oncologic selection criteria for this purpose were described in 1996 by Mazzaferro et al., with now >30 difference selection criteria described relying on a combination of traditional morphometric features and alpha feto protein (AFP) [54,55,56,57,58,59,60,61,62]. AFP has very well-described limitations as a biomarker for HCC [63]. Wehrle et al. have described general equivalence in oncologic outcomes with less restrictive selection criteria; however, this highlights that there is limited utility in further pursuing improvements to selection criteria without introducing more advanced precision oncology methods [64]. Furthermore, there are new ground-breaking therapies in HCC introduced with the rise in immunotherapy, yet there is a substantial proportion of patients who are IO non-responders, for which advanced radiomic and/or serologic biomarkers are desperately needed to guide therapy [65,66,67].

Increasing PET-MTV has been preliminarily associated with reduced survival in unresectable HCC treated with PD-1-/PD-L1-based immunotherapies [68]. MTV has also been described as predicting response to trans-arterial locoregional therapies, and as a prognostic factor in other LRT and surgical therapies as well [69,70]. Indeed, perhaps most excitingly, MTV on PET has been described as correlating with microvascular invasion on explanted specimens in liver transplantation, which is, in turn, one of the most significant negative prognostic factors after transplantation [71,72,73,74].

Finally, we hypothesize that the potential combination of MTV again with ctDNA and other precision oncology markers may provide the next stage in selection criteria, as these have both been demonstrated as markers of recurrence and increased disease burden [75,76].

As previously discussed, the above-cited studies are preliminary in nature. While they highlight the exciting potential for MTV use in HCC and LT, they are primarily retrospective studies with small sample sizes, which again impacts the generalizability of these results. Interestingly, HCC, from a pathophysiologic standpoint, is a relatively homogenous disease as compared to other solid-organ malignancies, so limitations with MTV applicability with respect to inter-tumor heterogeneity are reduced. However, important considerations must be undertaken to create equitable use of MTV in LT patients. Similar to work carried out in CRC regarding the evaluation of MTV with respect to region, race, ethnicity, and gender, future prospective studies assessing the impact of socioeconomic factors on MTV in HCC and LT are needed, given than integration of MTV in LT has significant clinical and psychosocial implications. As such, we propose the urgent need for combined radiomic and serologic biomarker research based on the preliminary data described above, with a potential new age in disease staging.

## 8. Important Considerations and Limitations

The role of MTV has grown significantly in the past two decades. While there is significant data regarding the use of MTV across various solid-tumor malignancies, it is very important to acknowledge the current limitations. Most critical is the lack of prospective and/or randomized data for most disease states, with most of the research currently representing exploratory studies. Furthermore, there exists variability in terms of measurement methods related to both programming and operator differences. Differences in the segmentation method can impact MTV calculation, which can in turn precipitate varying prognostic outcomes [19]. Additionally, there is a lack of standardized cut-off values for MTV which has limited adaptability in the clinical setting [77].

Reproducibility and reliability are paramount to obtaining generalizable results and creating standardized cut-off values, for which well-designed prospective studies are needed. Inter-scan variability is another limiting factor, where substantial variations in MTV measurements have been noted across serial PET scans, which can impact the consistency of MTV results [78]. Lastly, as mentioned before, tumor heterogeneity, including tumor biology, morphology, and metabolic activity across different regions of the same tumor (which can be seen in several malignancies such as head and neck cancer, breast cancer, and colorectal cancer), can impact MTV measurements [18,26,44]. As detailed throughout this review, even within certain clusters of oncologic diseases, there remains inter-tumor and even intra-tumor heterogeneity that impacts the ability to conduct large prospective studies that control for such variable factors. For instance, in HNC, the variation in tissue characteristics has impacted MTV values with respect to prognostication [16]. In breast cancer, genetic and histopathologic heterogeneity have made it difficult to design studies to account for inherent variability when studying these volumetric measures. Even in esophageal cancer, where there is more data available on the use of MTV, differences in EAC and ESCC have questioned the broad applicability of MTV values and cut-offs derived from these studies. Early studies in CRC have also suggested that changes in genetic makeup may impact MTV [44].

The presence of such tissue- and cancer-specific variability, even in the early studies cited in this narrative review, demonstrate the challenges that remain in creating protocolized MTV cut-off values that help inform prognostication and treatment response. A combination of translational and prospective clinical research is needed to develop a robust wealth of data that will allow for the in-depth characterization of MTV measures with respect to tissue-specific characteristics. Such research will perhaps serve as a steppingstone for broadening the applicability of MTV in solid-tumor malignancies, with the aim of standardizing its use in both clinical and research settings. We view it as unlikely that the same cut-offs for MTV will apply across the spectrum of morphologically different solid tumors. However, both within and across disease sites, validation is needed. For example, even within CRC/CRLM, the two studies of MTV in LT for CRLM found optimal cut-offs of 1.6 cm^3^ and 70 cm^3^ respectively, demonstrating why the standardized assessment of optimal cut-offs is critical [46,47,48].

In essence, the aforementioned limitations can add layers of complications that can make the comparison and standardization of MTV challenging. As such, it is crucial to consider these limitations when working towards expanding the utility of MTV in both research and clinical settings.

## 9. Conclusions

FDG-PET remains a critical imaging modality in the pretreatment and ongoing evaluation of patients with solid-tumor malignancies. Semi-quantitative parameters derived from PET-FDG such as metabolic tumor volume (MTV) have gained popularity in the last two decades, with an evidence-based rise in clinical use for management for several patient populations. Broadly speaking, MTV has shown to be predictive of certain patient outcomes, including OS and DFS, across several various solid-tumor malignancies. The ability to assess metabolically active disease portends significant clinical benefit, especially when risk stratifying patients and assessing treatment response. While there is growing body of literature to support the utilization of MTV, special considerations for cancer-specific features and further prospective validation studies are needed to develop robust data to optimize MTV applications across tumor types. Expanding the role of MTV in clinical management has limitless potential, especially with the rapid evolution, adaption, and integration of technology in healthcare.

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
