# Peer review of "PET-Assessed Metabolic Tumor Volume Across the Spectrum of Solid-Organ Malignancies: A Review of the Literature"

_biomedicines, 2025, doi:10.3390/biomedicines13010123_

Round 1

Reviewer 1 Report (Previous Reviewer 1)

Comments and Suggestions for Authors

The new version of the manuscript has been prepared according the revision of a different reviewer. I agree with the paragraphes insterted in the new version, even if my opinion about the manuscript was already positive; surely now the manuscript has been improved. 

Reviewer 2 Report (Previous Reviewer 2)

Comments and Suggestions for Authors

It is worth publication.

This manuscript is a resubmission of an earlier submission. The following is a list of the peer review reports and author responses from that submission.

Round 1

Reviewer 1 Report

Comments and Suggestions for Authors

The review is about the role of the parameter MTV in solid malignancies. I think that the update about the PET semiquantitative parameters is a very interesting topic in this moment, considering the advent of radiomic parameters, that seems to be valid but currently cannot be yet applied.

The manuscript makes an update about the MTV, and even if the method of the bibliographic research are not described (the manuscript is a descriptive review), the references included are appropriate and the manuscript organization is appropriate and very useful to read with important results. I appreciated the distinction in paragraphs that focalize on the most frequent solid malignancies and I really appreciated the figure that make a good point on the topic and can be considered a useful take-home message.

Author Response

Comment 1: The review is about the role of the parameter MTV in solid malignancies. I think that the update about the PET semiquantitative parameters is a very interesting topic in this moment, considering the advent of radiomic parameters, that seems to be valid but currently cannot be yet applied.

The manuscript makes an update about the MTV, and even if the method of the bibliographic research are not described (the manuscript is a descriptive review), the references included are appropriate and the manuscript organization is appropriate and very useful to read with important results. I appreciated the distinction in paragraphs that focalize on the most frequent solid malignancies and I really appreciated the figure that make a good point on the topic and can be considered a useful take-home message.

Thank you so much for your review and for pointing this out. We agree with your comment about this being a narrative/descriptive review. To help clarify our search methods, we included a brief sentence describing the search method in the last portion of the introduction on page 3 lines 120-123.

Reviewer 2 Report

Comments and Suggestions for Authors

This review focuses on the role of PET MTV (metabolic tumor volume) in several solid organ malignancies. While numerous studies have explored the utility of PET MTV in malignancies, it has yet to become a mandatory tool in clinical practice.

One notable limitation is the variability in the cut-off values for MTV used to distinguish prognosis. These values differ significantly across studies, primarily because they are derived from cohorts with diverse backgrounds, often characterized by small sample sizes, selection bias, or retrospective designs.

In each section, the author bases their conclusions primarily on one or two studies. This raises questions about whether the presented findings are robust enough to support definitive conclusions. Additionally, the influence of cancer tissue type on MTV values and outcomes introduces further variability, which the authors should address. A more nuanced discussion considering the heterogeneity of results across tissue types and study designs would be requested.

Author Response

Comment #1: This review focuses on the role of PET MTV (metabolic tumor volume) in several solid organ malignancies. While numerous studies have explored the utility of PET MTV in malignancies, it has yet to become a mandatory tool in clinical practice.

One notable limitation is the variability in the cut-off values for MTV used to distinguish prognosis. These values differ significantly across studies, primarily because they are derived from cohorts with diverse backgrounds, often characterized by small sample sizes, selection bias, or retrospective designs. In each section, the author bases their conclusions primarily on one or two studies. This raises questions about whether the presented findings are robust enough to support definitive conclusions.

Thank you so much for your review and for pointing this out. We agree with your comment. Therefore, we have included the limitations of the quality of available evidence cited in each section for the different oncologic subgroups that hinders wide applicability of MTV use in the current state. These changes were made throughout the text with the following corresponding page and line numbers: pages 3-4 lines 136-137, page 4 lines 159-171, page 5 line 188, page 5 lines 197-204, page 5 line 205, page 6 lines 242-253, page 6 line 271, page 6 line 273, page 7 lines 303-314, page 8 lines 353-365, and page 9 lines 396-405.

Comment #2: Additionally, the influence of cancer tissue type on MTV values and outcomes introduces further variability, which the authors should address. A more nuanced discussion considering the heterogeneity of results across tissue types and study designs would be requested.

Thank you so much for pointing this out. We agree with your comment. Therefore, we have revised our limitations section to include the impact of tissue specific characteristics that introduce further variability in MTV use and outcomes on pages 9-10 lines 428-452. The discussing on the study designs serving as an important limitation as highlighted in the aforementioned page/line numbers as per your guidance.
